

# Measurement report: A one-year study to estimate maritime contributions to PM₁₀ in a coastal area in Northern France

Frédéric Ledoux[1,★], Cloé Roche[1,★], Gilles Delmaire[2], Gilles Roussel[2], Olivier Favez[3], Marc Fadel[1,*], Dominique Courcot[1]

[1]Unité de Chimie Environnementale et Interactions sur le Vivant, UCEIV UR4492, FR CNRS 3417, Université du Littoral Côte d′Opale (ULCO), Dunkerque, France.
[2]Laboratoire d'Informatique Signal et Image de la Côte d'Opale, LISIC UR4491, Université du Littoral Côte d′Opale (ULCO), Calais, France.
[3]Institut National de l'Environnement Industriel et des Risques, INERIS, Parc Technologique ALATA, Verneuil-en-Halatte, France.

*Correspondence to*: Marc FADEL (marc.fadel@univ-littoral.fr)

[★] **These authors contributed equally to this work.**

**Abstract.** This work is focused on filling the lack of knowledge associated with natural and anthropogenic marine emissions on PM₁₀ concentrations in Northern France. For this purpose, a one-year measurement and sampling campaign for PM₁₀ has been performed at a French coastal site situated in front of the Straits of Dover. The characterization of PM₁₀ samples was performed considering major and trace elements, water-soluble ions, organic carbon (OC), elemental carbon (EC), and organic markers of biomass burning and primary biogenic emissions. Furthermore, the source apportionment of PM₁₀ was achieved using the constrained weighted-non-negative matrix factorization (CW-NMF) model. The annual average PM₁₀ was 24.3 µg/m³ with six species contributing to 69% of its mass (NO₃⁻, OC, SO₄²⁻, Cl⁻, Na⁺, and NH₄⁺). The source apportionment of PM₁₀ led to the identification of 10 sources. Fresh and aged sea-salts contributed to 37% of PM₁₀, while secondary nitrate and sulfate contributed 41%, biomass burning 10%, and Heavy Fuel Oil (HFO) combustion from shipping emissions contributed 5%, on yearly averages. Additionally, monthly evolution of the sources' contribution evidenced different behaviors with high contributions of secondary nitrate and biomass burning during winter. In the summer season, 10 times higher concentrations for HFO combustion (July compared to January) and the predominance of aged sea-salts versus fresh sea-salts were observed. Constant weighted trajectories showed that the sources contributing to more than 80% of PM₁₀ at Cape Gris-Nez are of regional and/or long-range origins with the North Sea and the English Channel as hotspots for natural and anthropogenic marine emissions and Belgium, the Netherlands, and the West of Germany as hotspots for secondary inorganic aerosols.

## 1 Introduction

Maritime transport is considered nowadays as a crucial transportation system used to ship goods and people over long distances. Due to globalization of production processes and the growth of global trade, this mean of transportation has been



increasing with over 80% of the volume of international trade of goods carried by sea in the last decades (Marmer et al., 2009; UNCTAD, 2021). In the meantime, shipping is known as a significant atmospheric source of pollutants especially near harbors and surrounding coastal areas (Contini and Merico, 2021; Jonson et al., 2020; Ausmeel et al., 2020). It is an important source of carbon monoxide (CO), nitrogen oxides ($NO_x$), sulfur oxides ($SO_x$), volatile organic compounds (VOCs), and particulate matter (Lv et al., 2018; Seppälä et al., 2021). In particular, several studies have highlighted the impact of shipping emissions on human health (Andersson et al., 2009; Corbett et al., 2007) since most of their emissions are estimated to occur within 400 km of land (Endresen et al., 2003; Jutterström et al., 2021). According to Zhang et al. (2021), 94 200 premature deaths worldwide were associated with particulate matter (PM) exposure due to maritime shipping in 2015.

Over the last decades, European countries have made large efforts to reduce emissions from several anthropogenic sources such as road traffic, industrial, power generation, etc. which led to the indirect increase of the contribution of shipping to the overall anthropogenic emissions (Viana et al., 2014). In order to reduce shipping emissions, different coastal areas (The North Sea, the English Channel, the Baltic Sea, etc.) have been classified as Sulfur Emission Control Areas (SECA) (EEA, 2013). In the latter areas, the sulfur content in marine fuels was limited to 0.1% starting 2015, after it was 1% between 2010 and 2015 and 1.5% before 2010 (Tang et al., 2020). The European Union has set a limit of 0.1% sulfur content fuel on ships at berth at EU ports since 2010 (EEA, 2021).

Several papers in the literature have estimated the contributions from shipping emissions (combustion of HFO) to air quality across Europe and were mainly focused on coastal areas of Italy, Spain, and Ireland (Mazzei et al., 2008; Viana et al., 2009; Pandolfi et al., 2011; Becagli et al., 2012; Contini et al., 2011; Hellebust et al., 2010; Donateo et al., 2014; Viana et al., 2014; Cesari et al., 2014; Bove et al., 2016; Amato et al., 2009). The annual mean contributions to PM in these studies varied between 1 and 12%. However, there were no studies focusing on the maritime contribution to PM in coastal areas of France and more specifically in the southern North Sea region that encompasses large harbours in different areas such as in Dunkerque, Calais, Rotterdam, Antwerp, etc.(EEA, 2013).

In Northern France, studies focusing on the characterization of $PM_{10}$ and the identification of its emission sources were conducted in urban and industrial areas (Crenn et al., 2015; Kfoury et al., 2016; Waked et al., 2014; Rimetz-Planchon et al., 2008; Ledoux et al., 2017). However, coastline sites can also be impacted by high particulate atmospheric background levels even without any direct influence from urban and/or industrial sources. On one hand, this might be due to the long-range transport influence as well as the gas-to-particle conversion (Waked et al., 2014; Matthias et al., 2010). On the other hand, there is still a lack of information regarding the impact of natural maritime emissions such as sea-salts (Manders et al., 2010) and anthropogenic ones such as marine traffic. It is worth noting that the southern North Sea is considered as one of the higher ship traffic density in the world (EEA, 2013). This is due to the fact that the Northern France region is bordered by the North-Sea and the English Channel that form together with the Straits of Dover a narrow corridor leading to one of the highest shipping concentration in the world (EEA, 2013).



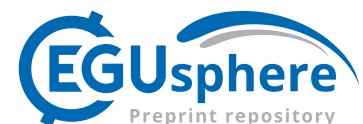

According to Ledoux et al. (2018), the impact of shipping emissions on $PM_{10}$ concentration in Calais urban area may reach up to about 30 µg/m³ on average, when the winds are blowing from the whole harbor area and compared to background concentrations. Punctually, ship emissions led to a marked increase of $PM_{10}$ mass of +78.9 µg/m³.

The main objective of this work is to understand the impact of emissions resulting from the maritime compartment in coastal areas in Northern France. Therefore, a one-year $PM_{10}$ sampling and measurement campaign was performed at the Cape Gris-

Nez, a French coastal site situated in front of the Straits of Dover in 2013. The collected samples were chemically characterized for their carbonaceous, ionic, and elemental fractions, as well as some organic tracers. Additionally, $PM_{10}$ sources were apportioned and studied, specifically natural emissions such as sea sprays and anthropogenic emissions linked to maritime traffic. This paper also highlights the seasonal variations of these $PM_{10}$ sources and the study of the regional influence. Even though the sampling campaign was conducted in 2013 and the regulations in SECA areas have evolved

since, this study is still relevant today due to the scarcity of literature studies in front of the Straits of Dover considered as one of the world's busiest shipping lanes. Additionally, despite the IMO regulation for global sulfur limit of 0.5% from ship's fuel oil applied starting January 2020, different countries are still adopting higher sulfur limits. It is worth noting that these limits were only set for sulfur content in marine fuels in order to reduce $SO_2$ emissions but no regulations for PM components neither in the sea nor at ports were issued. Hence, this study aims at highlighting the contribution of the natural

and anthropogenic marine emissions in the degradation of the air quality in a coastal background French site.

## 2 Materials and methods

### 2.1 Sampling site

The $PM_{10}$ filter sampling was conducted at the Cape Gris-Nez located in Northern France. It is a rural coastal site (50°52'08'' N, 1°35'49'' E, altitude 50 m), located on the edge of a cliff, 200 m from the sea and therefore strongly subject

to marine influence from sectors 210° to 50° via the north (**Figure 1**). The Strait of Dover, connecting the English Channel and the North Sea, is an intense navigation area with more than 500 boats passing every day (passage of oil tankers, merchant ships and fishing boats). It is considered as a strategic passage between Northern Europe and China, on one hand, and the Americas, on the other, through the ports of Antwerp, Rotterdam and Hamburg, representing nearly the quarter of the world's freight traffic.

The sampling site is far from major continental pollution sources and can be considered as a background site. The nearest town to the sampling site is Boulogne-sur-Mer, which is 16 km to the south, while Calais – a city that encompasses the fourth largest port in France – is about 21 km to the northeast. The A16 motorway is at a minimum distance of 10 km to the south-east. It is also worth noting that the study area is located in a SECA area where sulfur content was limited to 1% (between 2010 and 2015) for marine vessels including passenger ships and 0.1% for ships docked at the port since the

sampling was conducted in 2013.





## 2.2 Samples collection

PM$_{10}$ sampling and measurement campaign was carried out on 24-hour basis every three days from 1$^{st}$ of January to 31$^{st}$ of December 2013. During the sampling period, PM$_{10}$ concentrations were monitored using a MP101 beta gauge analyzer (Environment SA®). Moreover, PM$_{10}$ filters were collected using an automated high-volume sampler (DA80, DIGITEL®, Switzerland) operating at 30 m$^3$/h onto 150 mm Pall® QAT-UP filters. Filters were pre-heated for 4 hr at 450 ˚C before sampling to decrease the impurities. Over the sampling period, 122 samples have been collected and stored at -20 ˚C until analysis. Field blanks (two per month) were also considered by placing a blank filter in sampling conditions but without pumping. Additionally, meteorological data (temperature, wind speed and direction) were recorded on site using the WMT 52 ultrasonic wind sensor (Vaisala Windcap) coupled to the DIGITEL® DA80.

## 2.3 PM$_{10}$ chemical characterization

PM$_{10}$ chemical characterization included the analysis of the carbonaceous subfractions (OC and EC), major and trace elements, water-soluble ions, and some organic tracers. The detailed methods can be found in the supplementary information and will be briefly presented hereafter.

The analysis of OC and EC was done using a thermo-optical technique implementing the EUSAAR-2 protocol (Cavalli et al., 2010). Major and trace elements were analyzed following the protocol described in Ledoux et al. (2006) and Kfoury et al. (2016). Major elements (Al, Ba, Fe, Mn, P, Sr, Ti, and Zn) were analyzed by Inductively Coupled Plasma-Atomic Emission Spectrometry (ICP-AES), while trace elements (V, Cr, Ni, Sc, Co, Cu, As, Rb, Nb, Ag, Cd, Sn, Sb, Te, La, Ce, Tl, Pb, and Bi) were analyzed by ICP coupled to a mass spectrometer (ICP-MS). Cl$^-$, SO$_4^{2-}$, NO$_3^-$, Ca$^{2+}$, Mg$^{2+}$, K$^+$, Na$^+$, and NH$_4^+$ were analyzed by liquid ion chromatography following the protocol detailed in Ledoux et al. (2006) and Fadel et al. (2022). Finally, the analysis of organic compounds included the characterization of anhydrosugars (levoglucosan, mannosan, and galactosan), sugar alcohols (arabitol and mannitol), and monosaccharides (glucose and mannose) by High Performance Liquid Chromatography (HPLC) coupled to a Pulsed Amperometric Detector (PAD) (Srivastava et al., 2018).

## 2.4 Data analysis

### 2.4.1 Constrained weighted-non-negative matrix factorization (CW-NMF)

A Constrained Weighted Non-Negative Factorization (CW-NMF) model was applied in this study in order to identify and quantify the contribution of the sources to PM$_{10}$ concentrations. This model, developed by the LISIC (Laboratoire d'Informatique Signal et Image de la Côte d'Opale) at the University of Littoral Côte d'Opale, has the same principles as the USEPA PMF (Delmaire et al., 2010; Kfoury, 2013; Limem et al., 2014). Moreover, the model was adjusted in order to guide it in its calculations by adding constraints. The constraints added to this model are of two types: the "equality" constraints



that defines the presence or the absence of an element in the profile and the "boundary" constraints that impose a wide range of values in which the optimal solution can be found. These constraints were added in order to consider the "a priori"
knowledge of the chemical composition of the sources. For example, the concentration of levoglucosan was set zero in all the profiles except for the biomass burning source. Further details regarding the model can be found in Ledoux et al. (2017), and Kfoury et al. (2016).

The input data of the model consisted of the concentrations of 28 species in the 122 samples as well as their uncertainties. The chosen species were OC, EC, water-soluble ions ($Cl^-$, $SO_4^{2-}$, $NO_3^-$, $Ca^{2+}$, $Mg^{2+}$, $K^+$, $Na^+$, and $NH_4^+$), elements (Al, Cr, Fe,
Mn, P, Sr, Ti, Zn, V, Ni, Co, Cu, Cd, Sb, La, and Pb), and organic tracers namely levoglucosan for biomass burning and the sum of the concentrations of sugar alcohols and monosaccharides (named polyols) as tracers for primary biogenic emissions. Concerning the uncertainties, it was calculated by summing errors related to the analytical and sampling procedures. For trace elements, the calculated uncertainty was majored of 10% as it is usually reported (Prendes et al., 1999). In the case of species concentration below the detection limit (D.L.), it was replaced by D.L./2 and given an uncertainty of 100%. As for
missing data, it was replaced by the mean value and was assigned an uncertainty of 400% (Polissar et al., 1998; Kim et al., 2004). Additionally, the stability of the solution was evaluated by bootstrap analysis. It is based on the same principles of the bootstrap analysis used in EPA PMF 5.0. The profile of a source was validated if more than 80% of the bootstrap profiles were correlated (with $r^2 > 0.6$) to the reference one (the one obtained by considering the original dataset).

### 2.4.2 Conditional bivariate probability function (CBPF)


The conditional bivariate probability function (CBPF) combines wind speed with the conditional probability function (CPF). The latter is calculated as the probability that the concentration of a species, in a certain wind direction, is greater than a specific concentration value (which is the 75[th] percentile concentration of the different species in this study). CBPF will help to further understand the dependencies of the sources to the wind speed (Uria-Tellaetxe and Carslaw, 2014). These
representations were drawn using the open source software R and the open air package (Carslaw, 2015). The dataset used consisted of the species concentrations for each $PM_{10}$ sample for the 24h period distributed in front of the 48 corresponding wind speeds and directions (one measurement every 30 min). By that, the full dataset used to draw these representations count 122 x 48 lines.

### 2.4.3 Concentration weighted trajectories (CWT)


The Concentration Weighted Trajectories (CWT) approach investigates potential transport of pollutants and/or sources over large geographical scales (Polissar et al., 2001). It consists of combining species concentrations and/or source contributions with back-trajectories and use residence time information to identify air parcels that might be responsible of high concentrations observed at the receptor site (Petit et al., 2017).



Air mass back-trajectories were calculated using the HYSPLIT model considering the Gridded Meteorological Data Archives (GDAS 1 degrees). For each sample, eight 72h backward trajectories were assigned and the source contribution values for the 24 hours period were combined with the corresponding trajectories. These representations were achieved through the Zefir 4.0 software which is an Igor based package (Petit et al., 2017).

## 3. Results and discussions

**3.1 PM$_{10}$ concentration and composition**

The concentrations of PM$_{10}$ as well as the concentrations of OC, EC, water-soluble ions, and elements are reported in **Table 1** and the concentrations of the analyzed organic tracers are reported in **Table 2**. The yearly-mean PM$_{10}$ concentration obtained for the set of sampling days was of 24.3 µg/m$^3$ which is higher than the WHO PM$_{10}$ annual guideline value of 20 µg/m$^3$ that was applicable in 2013 (WHO, 2006) and the newly annual guideline value of 15 µg/m$^3$ (WHO, 2021). Even

though it is a rural background site, the PM$_{10}$ average concentration found at Cape Gris-Nez (CGN) was similar to the ones reported for urban and industrial sites in the Northern region of France in 2013 (24 µg/m$^3$) (Atmo, 2013), suggesting that PM$_{10}$ sources were mainly of regional influence rather than a local one (Ledoux et al., 2018).

The predominant chemical species were OC, Cl$^-$, Na$^+$, and secondary inorganic ions (NO$_3^-$, SO$_4^{2-}$, and NH$_4^+$), accounting for 69% of the average PM$_{10}$ concentrations **(Table 1)**. Secondary inorganic ions are found in the atmosphere due to the gas to

particle conversion of their corresponding precursors (NO$_x$, SO$_2$, and NH$_3$) emitted by different anthropogenic activities. The neutralization ratio between NH$_4^+$ and the sum of SO$_4^{2-}$ and NO$_3^-$ close to 1 shows that ammonium is predominately found in the atmosphere as ammonium sulfate and ammonium nitrate. The evaluation of the concentration ratios between SO$_4^{2-}$ and NH$_4^+$ on one hand and SO$_4^{2-}$ + NO$_3^-$ and NH$_4^+$ on the other hand shows that ammonium nitrate (67%) is approximately as twice as abundant compared to ammonium sulfate (33%).

Na$^+$ and Cl$^-$ are typical seawater components and highlight the importance of the marine influence on air quality at the investigated site. Most of the other analyzed species contribute to less than 0.1% of PM$_{10}$ average concentration with the exception of EC (1.3%), Mg$^{2+}$ (1%), Ca$^{2+}$ (0.9%), K$^+$ (0.6%), Al (0.3%), Fe (0.4%), and levoglucosan (0.2%) **(Table 1 and Table 2).**

OC and EC showed very strong correlation during the sampling period (r=0.89, p<0.05) meaning that these species were

mainly emitted from the same sources. According to the literature, OC-to-EC concentration ratios between 0.3 and 1 were reported for vehicles running on diesel, higher values were reported for biomass burning (3.4-14), while emissions from heavy fuel oil vessels show OC-to-EC ratios higher than 10 (Zhang et al., 2020; Moldanová et al., 2009; Fadel et al., 2022; Khan et al., 2021). The OC-to-EC ratio obtained in this study varied between 2.7 and 26.3 with an average ratio of 7.6. The high variability in the OC-to-EC ratio shows that these species are emitted from several sources and the interval found

highlights that the traffic exhaust emissions might not highly contribute to the emissions of carbonaceous matter at CGN.





The polar plot representations of OC and EC showed that the highest concentrations of carbonaceous matter were observed for winds blowing from the northeast (NE) and southeast (SE) sectors consisting mainly of continental winds **(Figures S1-a and S1-b)**. Furthermore, a pollution rose of the OC-to-EC ratio was done in order to understand if different sources of carbonaceous matter exist in the different wind sectors **(Figures S1-c)**. Without considering the wind speed parameter, the highest OC-to-EC ratios were observed in the southwest sector (values higher than 10) followed by the northeast (NE) and southeast (SE) sectors where values were between 6 and 10.

The evaluation of the polar plot representation of the OC-to-EC ratio **(Figures S1-d)** shows that ratios higher than 20 are observed in the southwest sector when wind speed is higher than 10 m/s **(Figure S1-d).** When examining the concentrations of OC and EC in these samples, the high OC-to-EC ratios was mainly due to the low EC concentrations ($< 200$ ng/m$^3$) found in these samples under maritime influence which increases the obtained values of the considered ratio. When removing these samples, OC-to-EC ratios were observed higher than 6 in the sectors northeast to southwest emphasizing on the influence of shipping emissions, biomass burning, and primary biogenic emissions **(Figure S1-e)**.

Anhydrosugars are mainly considered as tracers of biomass burning (Vincenti et al., 2022). These compounds show higher concentrations during the cold periods of the year. The average concentration of these compounds was 185 ng/m$^3$ between January and March 2013, 45 ng/m$^3$ between November and December, and 22 ng/m$^3$ for the rest of the year (warmer months). According to the literature, high levoglucosan-to-mannosan concentration ratio (close to 15) is indicative of hardwood combustion while low values (between 2 and 6) are mainly attributed to softwood combustion (Schmidl et al., 2008). The average ratio in this study is 7.3 which may be considered as indicative of a mix between the two types of wood.

Finally, the sugar alcohols and monosaccharides identified in this study are mainly of primary biogenic emissions. These compounds show at least three times higher concentrations during summer (June-August 2013) compared to the rest of the sampling days in 2013 (64 vs. 19 ng/m$^3$, respectively). Similar observations were reported by Samaké et al. (2019) in 28 French sites where the maximum concentrations were observed in the summer season. This might be linked to higher temperatures and humidity conditions in summer that leads to the growth and sporulation of fungal and prokaryotic cell activities (Samaké et al., 2019).

## 3.2 Source profiles

A progressive approach has been adopted in order to find the best solution by CW-NMF, consisting on increasing the number of factors and evaluating the profiles.

The best results were obtained for the 10 factors solution. The stability of the results was examined via bootstrap analysis and the different source profiles satisfied the validation criterion. Additionally, the reconstructed and observed PM$_{10}$ concentrations were very strongly correlated (slope of 0.95 and r$^2$=0.99) **(Figure S2).** Furthermore, the different species considered in the CW-NMF model were well reconstructed with slopes close to 1 and r$^2$ higher than 0.75.



The normalized profiles of the 10 identified sources at CGN are presented in **Figure 2.** The time series of the different profiles is presented in the supplementary data (**Figure S3**). Additionally, the distribution of the chemical species in the 10 identified sources is presented in **Figure S4**.

Two profiles related to sea-salts emissions were identified. The first profile shows the highest loading of $Na^+$ and $Cl^-$ between the profiles **(Figure S4)** with an average $Cl^-$-to-$Na^+$ ratio of 1.8 which is commonly observed for fresh sea-salts (Seinfeld and Pandis, 2016). The fresh sea-salts profile also included other ionic species such as $K^+$, $Mg^{2+}$, $Ca^{2+}$, and $SO_4^{2-}$. The second source profile was dominated by high contributions of $Na^+$ but no $Cl^-$ was found. This might be due to the $Cl^-$ depletion resulting from the reaction between sea-salts and $NO_x$ and $SO_2$ gas (Seinfeld and Pandis, 2006). $Cl^-$ was compensated with much higher contributions of $NO_3^-$ and $SO_4^{2-}$ in this profile compared to the fresh sea-salts as well as some elements from anthropogenic origins leading to the attribution of this profile to aged sea-salts.

The third profile contained a high proportion of Al, Fe, $Ca^{2+}$, and $K^+$ which are mainly the signature of crustal dust resuspension (Moreno et al., 2013). Two profiles show high loadings of secondary inorganic ions: the first one shows high loadings of $NO_3^-$ and $NH_4^+$ and was identified as "secondary nitrate" while the second one shows an abundance of $SO_4^{2-}$ and $NH_4^+$ and was ascribable to "secondary sulfate". The secondary nitrate profile also shows a considerable contribution of OC as well as some elements from anthropogenic origins. The presence of these species in the profile is mainly linked to the aging of the secondary inorganic aerosols and/or the effect of mixing with particles emitted from combustion sources (Koçak et al., 2015; Kfoury et al., 2016).

The following profile was attributed to biomass burning due to the presence of levoglucosan, OC, EC, as well as $K^+$, $SO_4^{2-}$, and $NO_3^-$ (Fadel et al., 2022). The levoglucosan-to-OC concentration ratio found in this profile (which is equal to 13) is within the range found in different studies for wood burning (between 7 and 17) (Fine et al., 2002). Additionally, the OC-to-EC ratio, which is found equal to 5.3, is also indicative of biomass burning emissions (Khan et al., 2021).

A profile attributed to road traffic was identified due to high loading of species emitted from the exhaust emissions (such as OC and EC) as well as elements emitted from non-exhaust emissions (such as Fe, Al, Zn, Cu, Sb). The OC-to-EC ratio of 0.7 is within the range of diesel traffic exhaust (0.3-1) (Salameh et al., 2018).

Heavy Fuel Oil (HFO) combustion profile was composed of high loadings of carbonaceous matter (OC and EC) as well as high contributions of V, Ni, $NO_3^-$, and $SO_4^{2-}$. The concentration of OC in this profile is 4.7 times higher than EC, which is consistent with other studies (Zhang et al., 2016). The evaluation of the V-to-Ni concentration ratio show a value of 1.3 in this study which is lower than the ones usually found (close to 3) from shipping emissions (Pandolfi et al., 2011; Becagli et al., 2012). Similar observations (V-to-Ni ratio of 1.6) were previously reported for other sites in the Northern region of France (Ledoux et al., 2017). This is mainly due to the position of the sampling sites reported in the latter study as well as the site of this study that are located in a SECA where sulfur content in marine fuels is limited to 1% during the sampling period in 2013, whereas the sites reported by Pandolfi et al. (2011) and Becagli et al. (2012) were not. The quality of the fuel with a low sulfur content used for shipping was shown to contain as well low contents of V and Ni and by that change their ratio (Zhang et al., 2016).



The primary biogenic emissions factor was identified due to the high contribution of carbonaceous matter as well as the sum of sugar alcohols and monosaccharides, considered as tracers of the source (Bauer et al., 2008). Additionally, we can observe a considerable contribution of P in the profile. It is known that one of the sources of atmospheric P might be the primary

biogenic ones (Shi et al., 2019).

Finally, the last profile shows high loadings of Al, Cr, Fe, Mn, and other elements such as Ni, Cu, Pb, La, V, Co, and Cd with neither ionic nor carbonaceous species. The profile was then named "metals-rich" and might be due to the industrial activities in the region (Ledoux et al., 2017). This factor explains more than 50% of Cr concentrations **(Figure S4)**.

**3.3 PM$_{10}$ source contributions**

The average contribution of the 10 sources to the PM$_{10}$ concentration at CGN during 2013 is presented in Figure 3. The secondary inorganic aerosols sources, namely secondary nitrate and secondary sulfate, contribute the most to the PM$_{10}$ concentration with a cumulative contribution of 41% (10 µg/m$^3$). The concentrations of secondary inorganic aerosols found in our study are higher than those found in Lens, an urban background site in Northern France, for a study in 2011-2012

(28% of PM$_{10}$, 5.9 µg/m$^3$) and in Nogent-sur-Oise, an urban site in Northern France in 2013 (27% of PM$_{10}$, 7.1 µg/m$^3$) (Oliveira, 2017; Waked et al., 2014).

The sea-salts (fresh and aged) contribute together to 37% of PM$_{10}$ concentration (9 µg/m$^3$) with higher contribution found to the fresh sea-salts compared to the aged ones. These contributions are found higher than those found for non-coastal sites in the Northern region such as in Rouen between 2010 and 2011 (21% of PM$_{10}$, 4.6 µg/m$^3$) and in Lens between 2011 and 2012

(8% of PM$_{10}$, 1.6 µg/m$^3$) (Waked et al., 2014; Favez et al., 2011) showing the high influence of sea spray emissions at the CGN site.

Biomass burning is also considered as an important source of PM$_{10}$ at CGN, contributing 10% (2.4 µg/m$^3$). This phenomenon is mainly observed in rural background sites where biomass, specifically wood, burning episodes are remarkably high (Golly et al., 2019). The contribution found in this study is similar to the yearly-average concentration

found in France for biomass burning estimated at 2.5 ± 1.2 µg/m$^3$ (Favez et al., 2021). This source is considered as the largest contributor to organic aerosols influencing the overall urban air quality in the country (Favez et al., 2021).

The five remaining sources contribute together to 12.3% of PM$_{10}$ **(Figure 3).** Due to the rural typology of the site, the road traffic contribution to PM$_{10}$ (2.2%, 0.42 µg/m$^3$) is lower than the ones found in Lens (6%, 1.2 µg/m$^3$) and in other urban sites in France (15 ± 7 % of PM$_{10}$) (Waked et al., 2014; Weber et al., 2021). It is important to mention that in CGN, the

contribution of HFO combustion that might be mainly related to shipping emissions (5.3%) is higher than the road traffic contribution. This percentage is in the range of the contribution from shipping emissions (1-7%) to PM$_{10}$ in different European coastal areas (Barcelona, Venice, Melilla, Algeciras, Lampedusa) (Pandolfi et al., 2011; Becagli et al., 2012; Viana et al., 2014; Amato et al., 2009; Viana et al., 2009). In addition to the contribution of shipping emissions to the PM$_{10}$ mass concentration, Viana et al. (2014) reported that this source may also influence new particle formation and thus further





contributing to the air quality degradation. It is worth noting that the contribution to $PM_{10}$ presented in this work corresponds to the direct emissions from the source. However, considerable amounts of $SO_2$ and $NO_x$ can be emitted from shipping activities and can be transformed into secondary compounds by gas-to-particle conversion that largely contribute to PM mass. Ledoux et al. (2018) have reported that the impact of shipping in the harbor of Calais in Northern France was estimated to 35% of NO, 51% of $SO_2$, and 15% of $NO_2$ average concentrations showing that the contribution of shipping emissions to $PM_{10}$ at CGN could be underestimated.

## 3.4 Monthly variations of the major $PM_{10}$ sources

The monthly variations of the contributions of the 10 identified sources at CGN during 2013 are presented in **Figure S5**. Secondary inorganic aerosols (secondary nitrate and sulfate) show a similar trend with higher contributions recorded during winter period. In this study, the highest contribution for ammonium nitrate was recorded in March (14 µg/m³ on average), when PM pollution episodes in Western Europe are dominated by secondary aerosols (Petit et al., 2019).

Secondary nitrate shows higher contributions during spring (March-May) and lower contributions during summer (June-August) compared to ammonium sulfate. This might be partially explained by the meteorological conditions and the semi-volatility of nitrate while ammonium sulfate is considered more thermally stable (Petit et al., 2019). Similar seasonal trends were observed by Waked et al. (2014) and Weber et al. (2019). Biomass burning shows a strong seasonality with the highest contributions during the cold period (January to march) due to wood burning for residential heating. The average contribution of biomass burning to $PM_{10}$ during these three months is 19.2% which is twice higher than the average contribution during the whole year.

Fresh and aged sea-salts as well as HFO combustion from shipping emissions, which contribute together to 42% of $PM_{10}$ during the total sampling period also show some temporal differences and were presented in **Figure 4.** The figure also presents their conditional bivariate probability function (CBPF) plots in order to further interpret the highest contributions observed for these sources. Different trends of contributions were observed for the fresh and aged sea-salts. The higher contributions of fresh sea-salts source were recorded in November (7 µg/m³) and in December (7.5 µg/m³) while those of aged sea-salts in April (4.5 µg/m³) and May (5.4 µg/m³). Additionally, the aged sea-salts contribution is found lower than the fresh-salts throughout the sampling period except for summer months where the average contribution concentrations of aged sea-salts were 2.3 and 2.8 µg/m³ compared to 2.1 and 0.8 µg/m³ for fresh sea-salts in July and August, respectively. The main reason for these differences might be the meteorological conditions. Indeed, the CBPF representation of fresh sea-salts clearly evidenced that the maximum concentrations were observed for winds blowing from the southwest (SW) and northeast (NE) wind sectors corresponding to the English Channel and the North Sea, respectively and for medium to high wind speeds (> 10 m/s) **(Figure 1)**. These wind directions were predominant during all months of the year except for the summer season which could explain the higher concentrations of fresh compared to aged sea-salts. On the other hand, the maximum concentrations of aged sea-salts were obtained when the wind blew from the northeast wind sector with wind





speeds higher than 10 m/s **(Figure 4)**. This might be explained by the reaction of the fresh sea-salts with $SO_2$ and $NO_2$ that also show the highest concentrations in the northeast wind sector **(Figure S6)** to yield aged sea-salts in the Strait of Dover

and the North Sea area **(Figure 1)**. The high concentrations of these gases in this wind sector are predominantly emitted by ships crossing the English Channel as suggested by Ledoux et al. (2018).

As for the HFO combustion from shipping emissions, the highest contribution was observed during July with ten times higher average concentration during this month (2.65 µg/m$^3$) compared to January (0.25 µg/m$^3$). This source contributes to 14% of $PM_{10}$ concentration during July which is approximately three times higher than the yearly average contribution

**(Figure 3)**. The CBPF representation shows that the high contribution concentrations of HFO combustion were observed for winds blowing from the northeast and southwest wind sectors with low wind speed (< 5 m/s). According to the Marine Management Organization (MMO), the number of ferries was 1.5 times higher during this specific period while the traffic in the English Channel – North Sea remains constant which could account for explaining the higher contribution (MMO, 2014). In addition to that, meteorological conditions characterized by low wet deposition and low wind speed favor the

accumulation of pollutants (Aulinger et al., 2015).

### 3.5 Regional influence

The representations of the source contributions with the air-mass back trajectories using the CWT method are useful to study the impact of regional emissions on $PM_{10}$ concentrations. The focus will be on the sources identified by CW-NMF that largely contribute to $PM_{10}$ concentrations and that can be of regional origins. For this purpose, we present in **Figure 5**, the

CWT representations for fresh and aged sea-salts, secondary nitrate and sulfate, as well as HFO combustion that correspond altogether to 83% of $PM_{10}$ during the sampling period at CGN **(Figure 3)**. The CWT analysis exhibits specific hotspots for sea-salts factors over the North Sea, and the English Channel. Additionally, the coastal part of the Atlantic Ocean can be considered as an additional hotspot for fresh sea-salts **(Figure 5)**. This can be also observed in the CBPF representations that show high contributions of the fresh sea-salts when the air blows from the southwest sector with high wind speeds (higher

than 10 m/s) **(Figure 4)**. The CWT model highlights once again the influence of HFO combustion corresponding to shipping emissions and shows that the North Sea, the English Channel, and the Strait of Dover can be considered as hotspots for this source.

On the other hand, secondary nitrate and secondary sulfate show similar geographical origins with main hotspots located over Belgium, the southern part of the Netherlands, and Western Germany.

These observations were consistent with other studies that highlighted that these sources are mainly inland and associated with continental air masses (Moufarrej et al., 2020; Petit et al., 2019). These regions were considered as important $SO_2$ emitters produced by power generation and transformation industries and $NO_2$ emitters from road transport, power plants, and other fuel converters (2012; Pay et al., 2010). Nevertheless, the North Sea can be also seen as a hotspot for these sources.



As for $PM_{10}$, the highest concentrations were traced back to marine areas such as the North Sea and the English Channel as well as some European countries such as Northern France, Belgium, Netherlands, and Western Germany. Waked et al. (2018) investigated the geographical origins of $PM_{10}$ impacting the North of France during the period between 2009 and 2013 and have also found similar results where air masses crossing Belgium, the Netherlands, and the North Sea were associated with intense anthropogenic activities and were considered as the highest potential source areas.


## 4 Conclusions

The main objective of this work was to determine the contribution of natural and anthropogenic maritime sources to $PM_{10}$ levels in a costal site in the North of France. In order to do that, a $PM_{10}$ sampling and measurement campaign was performed in 2013 at a coastal site in front of the Straits of Dover, named Cape Gris-Nez. $PM_{10}$ was chemically characterized for its

carbonaceous, elemental, and ionic fractions as well as some organic tracers of biomass burning and primary biogenic emissions. This exhaustive characterization was essential in order to explain the $PM_{10}$ concentration levels observed in the North of France and to estimate the sources' contributions to $PM_{10}$ using a receptor model which was Constrained Weighted Non-Negative Matrix Factorization.

In 2013, at Cape Gris-Nez, $PM_{10}$ mean value was 24.3 µg/m$^3$ and is very similar to those observed in several other sites in

Northern France. Six species explains about 70% of the total mass of $PM_{10}$: $NO_3^-$, OC, $SO_4^{2-}$, $Cl^-$, $Na^+$, and $NH_4^+$. Using the CW-NMF model, 10 source profiles were identified and were associated to natural and anthropogenic emissions. During the period of study, the mean contributions of the different sources were 37% for fresh and aged sea-salts, 41% for the secondary inorganic aerosols, 10% for biomass combustion, 5% for marine traffic. Additionally, the study of the monthly evolution of the sources' contribution shows that secondary nitrate and biomass burning were predominant during the cold season. As for

the summer season, the impact of marine traffic and the predominance of aged sea-salts versus fresh sea-salts were mainly evidenced. The contribution of HFO combustion from shipping in July was found to be ten times higher than in January.

Finally, CWT analysis showed that the North Sea, the English Channel, and the coastal part of the Atlantic Ocean are important hotspots for maritime emissions whereas Belgium, Netherlands, and the West of Germany are hotspots for secondary sources, emphasizing on the role of long-range transport on the air quality in the Cape Gris-Nez in particular and

in the North of France in general.

## Financial support

This project corresponds to a scientific contribution included in the Atmosphere Protection Plan funded by the Environment, Planning and Housing Regional Agency (Hauts-de-France) as well as by the French ministry of environment.

The "Unité de Chimie Environnementale et Interactions sur le Vivant", UCEIV UR4492, participates in the CLIMIBIO
project, which is financially supported by the Hauts-de-France Region Council, the Ministry of Higher Education and
Research, the European Regional Development Funds. Cloé Roche is grateful to the "Pôle Métropolitain Côte d'Opale"
(PMCO) for the funding of her PhD.

**Acknowledgment**

The authors would like to thank the "Centre Commun de Mesures, ULCO" and specifically Fabrice Cazier and Dorothée
Dewaele, as well as the regional air quality monitoring network Atmo Haut-de-France, for their contribution to this project.
The authors would also like to thank Amaury Kasprowiak for his help in the ionic chromatography analysis.

**Author contribution**

Frédéric Ledoux: data curation, conceptualization, supervision, investigation, review & editing, data analysis, and formal
analysis.
Cloé Roche: sampling, chemical analysis, investigation, data curation, visualization, formal analysis, investigation, and
writing.
Gilles Delmaire: Software, and formal analysis.
Gilles Roussel: Software, and formal analysis.
Olivier Favez: supervision of chemical analysis, review & editing.
Marc Fadel: data curation, visualization, writing original draft, review & editing.
Dominique Courcot: funding acquisition, conceptualization, supervision, investigation, formal analysis, project
administration, and review & editing.

**Competing interests**

The authors declare no conflict of interest or competing interests.

**Data availability**

Data used for this study can be found at https://doi.org/10.5281/zenodo.7664528.



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





**Table 1: Average, standard deviations (S.D.), minimum (Min.), and maximum (Max.) concentrations of PM$_{10}$ and its chemical components (OC, EC, water-soluble ions in µg/m$^3$ and elements in ng/m$^3$) at Cape Gris-Nez (CGN) during the sampling period in 2013.**

| Concentrations | | **Average** | **S.D.** | **Min.** | **Max.** |
|---|---|---|---|---|---|
| PM$_{10}$ (µg/m$^3$) | | 24.3 | 13.6 | 5.0 | 74.0 |
| Carbonaceous fraction (µg/m$^3$) | OC | 2.14 | 2.05 | 0.33 | 13.7 |
| | EC | 0.32 | 0.28 | 0.02 | 1.69 |
| **Total carbon (TC) (µg/m$^3$)** | | **2.45** | **2.33** | **0.35** | **15.4** |
| Water-soluble ions (µg/m$^3$) | NO$_3^-$ | 5.16 | 5.92 | 0.21 | 33.2 |
| | SO$_4^{2-}$ | 3.01 | 2.47 | 0.54 | 13.9 |
| | Cl$^-$ | 2.48 | 2.56 | 0.003 | 9.3 |
| | Na$^+$ | 2.11 | 1.52 | 0.10 | 7.88 |
| | NH$_4^+$ | 1.94 | 2.54 | 0.03 | 12.9 |
| | Mg$^{2+}$ | 0.25 | 0.18 | 0.02 | 0.89 |
| | Ca$^{2+}$ | 0.21 | 0.26 | 0.05 | 2.26 |
| | K$^+$ | 0.14 | 0.09 | 0.02 | 0.64 |
| **Total water-soluble ions (µg/m$^3$)** | | **15.3** | **10.2** | **4.25** | **56.4** |
| Elements (ng/m$^3$) | Fe | 104 | 130 | 0.42 | 19.7 |
| | Al | 74.6 | 85.8 | 0.68 | 584 |
| | P | 30.8 | 43.5 | 1.37 | 253 |
| | Zn | 14.6 | 21.6 | 0.06 | 168 |
| | V | 5.68 | 6.19 | 0.31 | 33.8 |
| | Ni | 4.69 | 5.12 | 0.07 | 26.9 |
| | Pb | 4.67 | 5.62 | 0.12 | 38.4 |
| | Ti | 4.61 | 6.23 | 0.04 | 32.5 |
| | Mn | 4.31 | 7.03 | 0.01 | 40.3 |
| | Sc | 2.77 | 2.76 | 0.21 | 15.9 |
| | Cu | 2.22 | 3.08 | 0.03 | 22.3 |
| | Ba | 1.85 | 3.13 | 0.06 | 19.7 |
| | Sr | 1.75 | 1.05 | 0.15 | 5.98 |
| | Cr | 1.06 | 1.06 | 0.21 | 6.01 |
| | Sn | 0.69 | 0.88 | 0.01 | 5.49 |
| | Sb | 0.59 | 0.62 | 0.01 | 3.84 |
| | As | 0.29 | 0.38 | <D.L. | 2.18 |
| | Rb | 0.29 | 0.34 | 0.02 | 1.90 |
| | Bi | 0.22 | 0.44 | <D.L. | 2.21 |
| | Te | 0.21 | 0.30 | <D.L. | 2.01 |
| | Co | 0.18 | 0.20 | 0.01 | 1.25 |
| | La | 0.17 | 0.13 | 0.002 | 0.60 |
| | Ce | 0.12 | 0.13 | 0.005 | 0.72 |
| | Cd | 0.11 | 0.15 | 0.001 | 0.81 |
| | Ag | 0.04 | 0.06 | 0.001 | 0.31 |
| | Tl | 0.02 | 0.05 | <D.L. | 0.31 |
| | Nb | 0.01 | 0.02 | <D.L. | 0.11 |
| **Total elements (ng/m$^3$)** | | **260** | **246** | **21.7** | **1434** |



**Table 2: Average, standard deviations (S.D.), minimum (Min.), and maximum (Max.) concentrations of organic tracer species (in ng/m³) in PM₁₀ at Cape Gris-Nez (CGN) during the sampling period in 2013.**


| Concentrations in ng/m$^3$ | | **Average** | **S.D.** | **Min.** | **Max.** |
|---|---|---|---|---|---|
| **Anhydrosugars** | levoglucosan | 55.2 | 110 | 0.01 | 853 |
| | mannosan | 7.60 | 12.2 | 0.01 | 90.6 |
| | galactosan | 3.18 | 7.16 | 0.01 | 51.4 |
| **Sugar alcohols** | arabitol | 10.7 | 25.1 | 0.01 | 232 |
| | mannitol | 10.7 | 20.0 | 0.01 | 184 |
| **Monosaccharides** | glucose | 6.71 | 9.89 | 0.01 | 60.0 |
| | mannose | 1.90 | 2.76 | 0.01 | 14.6 |





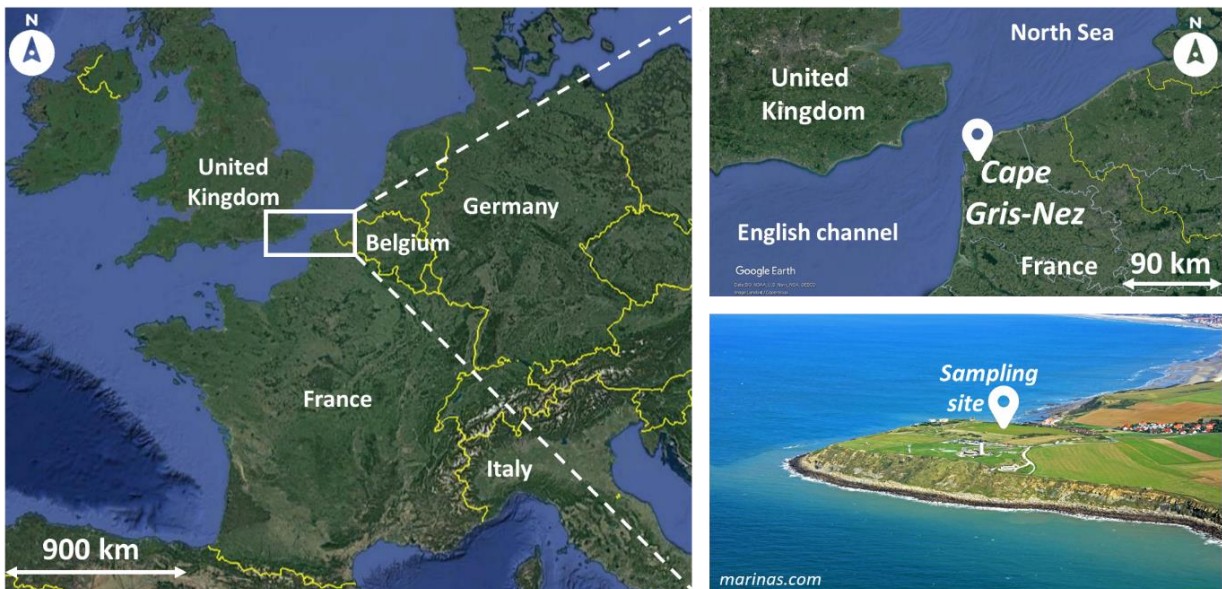

**Figure 1: Location of the sampling site (© Google Earth)**





**Figure 2: PM$_{10}$ source profiles at Cape Gris-Nez (CGN) identified using the CW-NMF model**





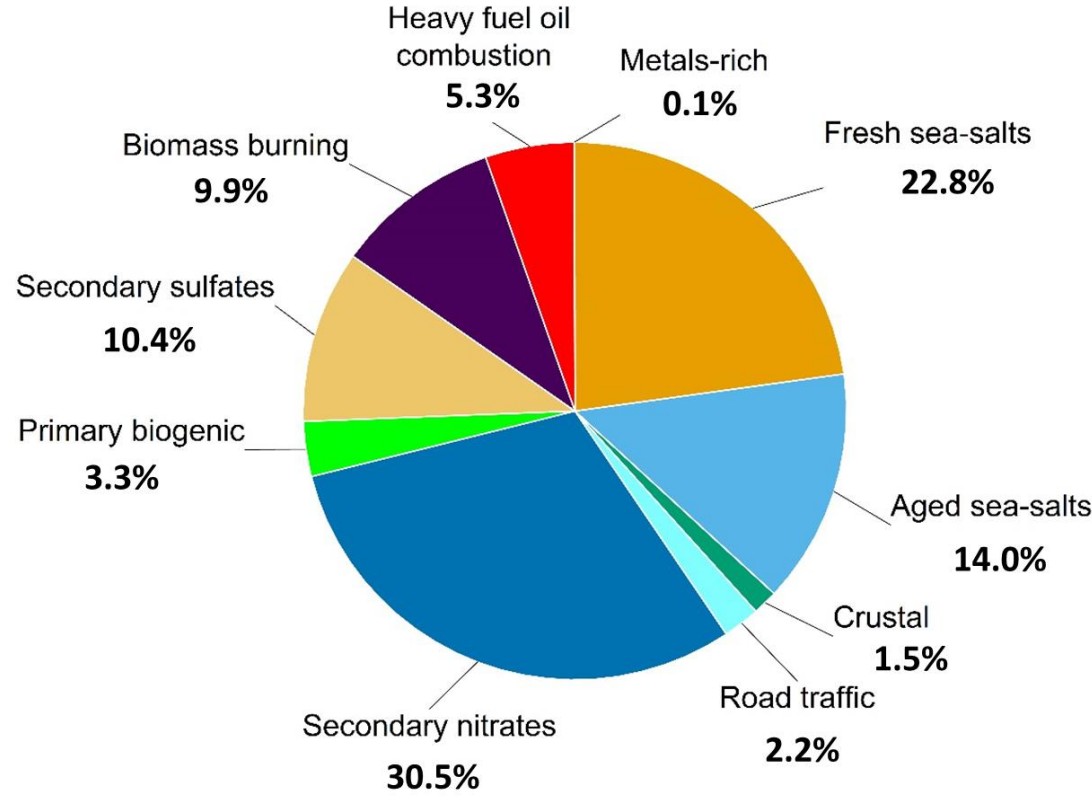

**Figure 3: Mean source contributions to PM₁₀ collected at Cape Gris-Nez (CGN) during 2013**








**Figure 4: Monthly average, median and 25th and 75th percentiles of the contributions (ng/m³) of the PM₁₀ maritime sources at CGN as well the conditional bivariate probability function (CBPF) plots**





**Figure 5: CWT results for PM₁₀ and some NMF factors (fresh sea-salts, aged sea-salts, secondary nitrate, secondary sulfate, and HFO combustion). Red colors highlight potential emission zones. Contribution scales are in µg/m³.**