# Peer review of "Measurement report: A one-year study to estimate maritime contributions to PM10 in a coastal area in Northern France"

_EGUsphere, 2023_

## Author Comment (AC1)

**Answers to referee # 1**

**General comment**

The paper reports a study of maritime contributions to $PM_{10}$ in a rural coastal area of northern France. The topic is of interest and suitable for the Journal. The approach used is quite standard, however, the extensive dataset, the detailed statistical analysis, and the limited availability of data for this area make the paper useful to the scientific community. I would like to see it published after a revision step addressing my minor specific comments.

*We would like to thank the referee for his constructive comments that aim at improving the manuscript. Hereafter, we have replied to the different comments in blue. Amendments in the manuscript will be done once the final version uploaded.*

**Specific comments**

1. Please remove all instances of etc. in the paper. If something should be added, please do it explicitly.

*The « etc. » term will be removed from lines 42, 44, and 54.*

2. Line 67. Please remove the "+". In addition, this value seems to be very large compared to the measured ones and to the values published in other studies. What it represents Annual average or other short-term contribution?

*The « + » symbol will be removed from line 67. This information was retrieved from Ledoux et al. (2018) that studied the influence of shipping emissions on PM concentrations in the port city of Calais, one of the busiest harbors in Europe. The sampling was conducted for three months and the sampling site was situated at a distance of 500 to 1000 meters from boats. $PM_{10}$ concentrations were deduced from the real-time particle size distribution measured by a Electrical Low Pressure Impactor. The temporal resolution for $PM_{10}$ was 1 min and 15 min for gas concentrations.*

*In order to estimate the impact of in-port ships on PM$_{10}$ concentrations, the authors have selected the data registered when the wind was blowing from the direction of the port. Periods showing no concentration peaks of PM, SO$_2$, NO, and NO$_2$ represented the background concentrations whereas periods for which successive peaks were detected represented the "Background + in-port shipping related concentrations". Results showed that the in-port ship emission increased the background PM$_{10}$ concentration by an average of 28.9 µg/m$^3$ with a maximum increase of 78.9 µg/m$^3$ for a 1-minute time interval (the value presented in line 67 of the manuscript).*

*The information will be modified in the manuscript as follows (Line 67 in the original manuscript):" Punctually, ship emissions led to a marked increase of PM$_{10}$ mass of +78.9 µg/m$^3$ for a one-minute time interval."*

3. Lines 77-80. It would be worth to mention here that the use of low-sulphur fuels actually reduce also primary PM emissions from ships how it has been demonstrated in several study and, of course, also secondary sulphate. So that the regulation was not done only for SO2.

*The authors agree with the referee. The information was added at the end of the introduction:*

*"Additionally, despite the IMO regulation for global sulfur limit of 0.5% from ship's fuel oil applied starting January 2020, different countries are still adopting higher sulfur limits. It is worth noting that these limits were set for sulfur content in marine fuels in order to reduce not only SO$_2$ emissions but also primary PM emissions (Shen and Li, 2020; Zetterdahl et al., 2016). However, no regulations for PM components neither in the sea nor at ports were issued. Hence, this study aims at highlighting the contribution of the natural and anthropogenic marine emissions in the degradation of the air quality in a coastal background French site."*

4. Line 101. Are these quartz filters?

*The Pall® QAT-UP filters are indeed no binder quartz filters. The sentence (lines 100-101) will be modified as follows:*

*"Moreover, $PM_{10}$ were collected using an automated high-volume sampler (DA80, DIGITEL®, Switzerland) operating at 30 $m^3/h$ onto 150 mm Pall Tissuquartz™ 2500 QAT-UP filters (no binder) filters."*

5. Line 172. Are you referring to secondary organic aerosol here?

*The authors were not referring to secondary organic aerosols. The interpretation found in this sentence was based on the comparison between the average concentration of $PM_{10}$ at Cape Gris-Nez which is a rural background site with no primary emission sources nearby and other urban and industrial sites in the Northern France region and during the same period (one year). $PM_{10}$ concentrations are mainly governed by the fluctuations of the background levels observed at a regional scale. Additionally, the long-range transport of different organic and inorganic species to Cape Gris-Nez have also an impact on the $PM_{10}$ concentration. This was also highlighted in the CWT representations showing that more than 80% of $PM_{10}$ concentrations was attributed to regional and/or long-range transport origins.*

6. Section 3.2. Considering that the receptor model used is still not widely applied, it should be useful to mention that it has comparable performances to the PMF model investigated in the recent work of Belis et al. (Atmospheric Environment: X, 5, 2020, 100053).

*The authors thank the referee for the suggestion. However, the "CW-NMF" model developed by the LISIC (Laboratoire d'Informatique Signal et Image de la Côte d'Opale) at the University of Littoral Côte d'Opale and that was used in this study was not mentioned in the recent work of Belis et al., 2020. For this reason, the reference will not be added in this particular context. However, it will be added in the introduction of the manuscript to highlight the importance of the source apportionment models.*

7. Lines 247-256. For the discussion on the V/Ni ratio, I suggest to have a look and mention the work of Gregoris et al (Environ Sci Pollut Res (2016) 23, 6951–6959) that shows ratios lower than expected in coastal areas with relevant impact of shipping as well as a strong spatial variability of this ratio.

*The authors would like to thank the referee for this suggestion. The reference of Gregoris et al. (2016) as well as the information within the paper related to the V/Ni concentration ratio will be added to the final manuscript as follows:*

*"Heavy Fuel Oil (HFO) combustion profile was composed of high loadings of carbonaceous matter (OC and EC) as well as high contributions of V, Ni, $NO_3^-$, and $SO_4^{2-}$. The concentration of OC in this profile is 4.7 times higher than EC, which is consistent with other studies (Zhang et al., 2016). The evaluation of the V-to-Ni concentration ratio show a value of 1.3 in this study which is lower than the ones usually found (close to 3) from shipping emissions (Becagli et al., 2012; Pandolfi et al., 2011). Similar observations (V-to-Ni ratio of 1.6) were previously reported for other sites in the Northern region of France (Ledoux et al., 2017). This is mainly due to the position of the sampling sites reported in the latter study as well as the site of this study that are located in a SECA where sulfur content in marine fuels is limited to 1% during the sampling period in 2013, whereas the sites reported by Pandolfi et al. (2011) and Becagli et al. (2012) were not. The quality of the fuel with a low sulfur content used for shipping was shown to contain as well low contents of V and Ni and by that change their ratio (Streibel et al., 2017; Zhang et al., 2016)."* **Gregoris et al. (2016) also found V/Ni concentration ratios that were less than 3 in Venice and the species were mainly attributed to HFO combustion from shipping.**

8. Lines 293-295. I would not say that it is underestimated. The pint is that it was analysed here only the contribution to primary PM10 and that there would be also a contribution to secondary aerosol, mainly sulphate, that could be even larger than the primary one.

*This work presented an exhaustive chemical characterization of $PM_{10}$ including primary species as well as secondary ones, namely the secondary inorganic aerosols ($NO_3^-$, $SO_4^{2-}$, and $NH_4^+$). The concentration levels of these secondary species characterized in $PM_{10}$ can be found in Table*

*1 of the manuscript. Additionally, these secondary species were added into the CW-NMF model in order to identify different $PM_{10}$ sources and quantify their contribution as well. The HFO combustion profile (Lines 247-256 in the original manuscript) was identified by high loadings of carbonaceous matter, V and Ni, as well as secondary inorganic species ($NO_3^-$ and $SO_4^{2-}$). By that, we cannot say that the HFO combustion source corresponds only to primary $PM_{10}$ emissions.*

*However, the authors wanted to highlight the idea that the transformation reaction of $SO_2$ emitted from the shipping emissions to $SO_4^{2-}$ is probably not at the equilibrium yet and maybe more amounts of $SO_4^{2-}$ were transformed which may lead to higher contribution of shipping emissions to $PM_{10}$. That is why the authors mentioned the underestimation of the shipping emissions contribution to primary $PM_{10}$.*

*The sentence in the original manuscript was modified as follows:*

*"However, considerable amounts of $SO_2$ and $NO_x$ can be emitted from shipping activities and can be transformed into secondary compounds by gas-to-particle conversion that largely contribute to PM mass. Ledoux et al. (2018) have reported that the impact of shipping in the harbor of Calais in Northern France was estimated to 35% of NO, 51% of $SO_2$, and 15% of $NO_2$ average concentrations.*

*However, the transformation reaction of $SO_2$ emitted from the shipping emissions to $SO_4^{2-}$ is probably not at the equilibrium yet and maybe a higher amount of $SO_4^{2-}$ could be formed from $SO_2$ which may lead to a probable underestimation of shipping emissions to primary $PM_{10}$."*

9. Lines 322-325. Please mention how and where these gases are measured.

*The information regarding the measurement of the gas concentrations will be added in the supplementary material:*

**"$SO_2$ and $NO_2$ were continuously measured using an AF21M $SO_2$ analyzer (Environnement SA, France) and an AC32M $NO$-$NO_2$-$NO_X$ analyzer (Environnement SA, France), respectively. The calibration of the analyzers was done at the beginning of the campaign and**

*were routinely check by the regional air quality network atmo Hauts-de-France. The temporal resolution was 15 minutes for gas concentrations. More details related to the measurement campaign of the gases at Cape Gris-Nez can be found in Ledoux et al. (2018)."*

10. Figure 4. Please use the same acronym as in the text (i.e. CBPF). To be honest, I do not understand why the CBPF of sea salt and aged sea salt are so different. A better interpretation of this aspect would be useful.

*The terms "CPF" in Figure 4 will be replaced by CBPF.*

*The difference between the CBPF of fresh and aged sea-salts is mainly due to the meteorological conditions as mentioned in the manuscript (Line 317). It is important to remember that the sampling site at Cape Gris-Nez is a rural coastal site, located at 200 m from the sea and therefore strongly subjected to fresh marine influence from sectors 210° to 50° via the north. This is why, for the fresh sea-salts, the maximum concentrations were observed for winds blowing from the southwest and the northeast sectors corresponding to the English Channel and the North Sea (for medium to high wind speeds > 10m/s).*

*As for the aged sea salts, the highest concentrations were observed for winds blowing from the northeast. In fact, the aged sea-salts are yielded by the reaction between the fresh sea-salts and the $SO_2$ and $NO_2$ gases that show the highest concentrations levels on land (Northeast wind sector – Figure S6). This phenomenon will be done according to the trajectory of the air masses. By that, the aged sea-salts may not come from the wind direction open to the sea but from land (Northeast wind sector) especially in a coastal site, which is the case of this study.*

11. Figure S4. The use of three different level of blue is not a good choice because make the figure hardly readable generating confusions among the different sources. Please make a different choice for these colours

*Following the referee's recommendation, various colors will be used for Figure S4 instead of the different levels of blue.*

References:

Becagli S, Sferlazzo DM, Pace G, di Sarra A, Bommarito C, Calzolai G, Ghedini C, Lucarelli F, Meloni D, Monteleone F, Severi M, Traversi R, Udisti R. Evidence for heavy fuel oil combustion aerosols from chemical analyses at the island of Lampedusa: a possible large role of ships emissions in the Mediterranean. Atmos. Chem. Phys. 2012; 12: 3479-3492.https://doi.org/10.5194/acp-12-3479-2012.

Gregoris E, Barbaro E, Morabito E, Toscano G, Donateo A, Cesari D, Contini D, Gambaro A. Impact of maritime traffic on polycyclic aromatic hydrocarbons, metals and particulate matter in Venice air. Environmental Science and Pollution Research 2016; 23: 6951-6959.https://doi.org/10.1007/s11356-015-5811-x.

Ledoux F, Kfoury A, Delmaire G, Roussel G, El Zein A, Courcot D. Contributions of local and regional anthropogenic sources of metals in $PM_{2.5}$ at an urban site in northern France. Chemosphere 2017; 181: 713-724.https://doi.org/10.1016/j.chemosphere.2017.04.128.

Ledoux F, Roche C, Cazier F, Beaugard C, Courcot D. Influence of ship emissions on $NO_x$, $SO_2$, $O_3$ and PM concentrations in a North-Sea harbor in France. J. Environ. Sci. 2018; 71: 56-66.https://doi.org/10.1016/j.jes.2018.03.030.

Pandolfi M, Gonzalez-Castanedo Y, Alastuey A, de la Rosa JD, Mantilla E, de la Campa AS, Querol X, Pey J, Amato F, Moreno T. Source apportionment of $PM_{10}$ and $PM_{2.5}$ at multiple sites in the strait of Gibraltar by PMF: impact of shipping emissions. Environ Sci Pollut Res Int 2011; 18: 260-9.https://doi.org/10.1007/s11356-010-0373-4.

Shen F, Li X. Effects of fuel types and fuel sulfur content on the characteristics of particulate emissions in marine low-speed diesel engine. Environmental Science and Pollution Research 2020; 27.10.1007/s11356-019-07168-6.

Streibel T, Schnelle-Kreis J, Czech H, Harndorf H, Jakobi G, Jokiniemi J, Karg E, Lintelmann J, Matuschek G, Michalke B, Müller L, Orasche J, Passig J, Radischat C, Rabe R, Reda A, Rüger C, Schwemer T, Sippula O, Stengel B, Sklorz M, Torvela T, Weggler B, Zimmermann R. Aerosol emissions of a ship diesel engine operated with diesel fuel or

heavy fuel oil. Environmental Science and Pollution Research 2017; 24: 10976-10991.10.1007/s11356-016-6724-z.

Zetterdahl M, Moldanová J, Pei X, Pathak RK, Demirdjian B. Impact of the 0.1% fuel sulfur content limit in SECA on particle and gaseous emissions from marine vessels. Atmos. Environ. 2016; 145: 338-345.https://doi.org/10.1016/j.atmosenv.2016.09.022.

Zhang F, Chen Y, Tian C, Lou D, Li J, Zhang G, Matthias V. Emission factors for gaseous and particulate pollutants from offshore diesel engine vessels in China. Atmos. Chem. Phys. 2016; 16: 6319-6334.https://doi.org/10.5194/acp-16-6319-2016.

---

## Author Comment (AC2)

**Answers to referee # 2**

**General comment**

The manuscript reports a standard source apportionment study with a focus on the maritime source. Beyond the original results and the use of a specific receptor site to achieve the goals, the rest of the study presents only the trivial application of well-established source apportionment techniques (PMF-like, CWT, CPF). Another weakness of this study is the "age" of the data, which were collected 10 years ago. However, since the lack of papers focusing on maritime sources, I am in favour of publication. The authors are requested to improve the manuscript by adding more information about the outcomes of the CW-NMF model (in particular about the number of factors extracted).

*We would like to thank the referee for his constructive comments that aim at improving the manuscript. We understand the referee's point of view regarding the outcomes of the CW-NMF model and the "overfitting" problem addressed in the following comments. We have checked again the reconstruction of the species added into the model and came to the conclusion that three elements (La, Mn, and Cr) were not as well reconstructed as the other species. For that purpose, we had run again the CW-NMF by removing these elements. In this case, the optimal number of factors obtained was nine, removing by that the "metal-rich" source. The outputs of the CW-NMF model specifically the reconstruction of $PM_{10}$ as well as the different species and the detailed boostrap analysis that show the robustness of the 9-factor solution will be presented in details in the following comments and will be added in the supplementary information. Amendments in the manuscript especially regarding the outcomes of the CW-NMF model have been done.*

**Specific comments**

1. Lines 76-79 "Additionally, despite the IMO regulation for global sulfur limit of 0.5% from ship's fuel oil applied starting January 2020, different countries are still adopting higher sulfur limits. It is worth noting that these limits were only set for sulfur content in marine fuels in order to reduce SO2 emissions but no regulations for PM components neither in the sea nor at ports were issued". Not clear why this consideration is reported within the objectives of the study. Is this topic addressed in the current study? Please clarify.

*The authors understand the point of view of the referee. This topic is not addressed in the current study. However, it was relevant to include it in this part of the introduction in order to shed the light on the idea that the data is still relevant nowadays since different countries are still adopting sulfur limits higher than 0.5%, which was the case in 2013. Additionally, it was added following the recommendation of the referee #1.*

2. Lines 66-80. Please comment on the choice of sampling PM2.5 to quantify the maritime contribution. Combustion emissions are expected to be finer than other sources. On the contrary, other sources in the coastal areas (e.g., sea salt) may emit large particles. Thus, it would be better to analyze PM2.5. Please comment to support your choice.

*We agree with the referee regarding the fact that combustion emissions are expected to be in the finer fraction of PM such as $PM_{2.5}$. However, the study was conducted in 2013. Several years before, the EU has placed policies in order to reduce the concentrations of some atmospheric pollutants. The directive 2008/50/EC limits daily $PM_{10}$ concentrations to 50 $\mu g/m^3$ with a maximum of 35 days of exceedance per year. This directive is binding and forces countries that do not comply with it to seek solutions for improvement. This was particularly the case of France, which was in a dispute on this subject with the European Union. Several regions in France were thus concerned with high number of exceedance days, especially the Northern region. This is why it was important at the time to identify the sources that contribute to the concentrations of $PM_{10}$ in order to focus on the reduction of emissions of these sources and to quantify the part of maritime contribution.*

3. Line 90. "The sampling site is far from major continental pollution sources and can be considered as a background site." Which background? Rural, urban, suburban? If rural, are you sure the site has the characteristics to be classified in this way? Please comment.

*We agree with the comment of the referee regarding the classification of the site as "background". According to the current Implementing Provisions on Reporting (IPR) 2011/850/EC regarding the type of sampling area and the influence of the immediate surroundings (traffic, industrial or*

*background), the site cannot be considered as a background site due to its coastal typology with marine anthropogenic emissions (https://www.eionet.europa.eu/aqportal/doc/IPR%20guidance_2.0.1_final.pdf). The term "background" will be removed from the manuscript. As for the definition of a rural site: it corresponds to an area that does not fulfill the criteria of an urban or suburban site with at least 10 km from major sources. The sampling site can be indeed considered as a rural site on the coast.*

4. Section 2.2. The sampling campaign is 1 year long (1st of January to 31st of December 2013) and samples were collected at daily frequency, however only 122 samples were analyzed. Why? How the samples to be analyzed have been selected? Randomly? 1 day over 3? Please explain.

*The sampling campaign was conducted between the 1ˢᵗ of January and the 31ˢᵗ of December 2013. A total number of 362 samples were collected at Cape Gris-Nez in 2013. 122 samples were chosen for the chemical analysis. They were not chosen randomly and they represent a sampling of 1 day over 3 in order to ensure that these samples are representative of the whole sampling period.*

*The information was added in the manuscript:*

*"Over the sampling period, 362 samples have been collected. 122 samples representing a sampling of one day over three were chosen for the chemical analysis and were presented in this study."*

5. Line 100. Was the beta monitor tested against the gravimetric measurement? Which calibration audit? Please add info.

*The beta monitor results were not compared to the gravimetric measurement of the filters. However, the LCSQA (Laboratoire central de la surveillance de la qualité de l'air) in France has verified the equivalence of the MP101M-RST measurements using the beta gauge with the*

*gravimetric reference method in their study conducted in 2011 and published in 2012 (LCSQA, 2012).*

*The information will be added in the final version of the manuscript as follows (Line 100): "During the sampling period, $PM_{10}$ concentrations were monitored using a MP101 beta gauge analyzer (Environment SA®). The analyzer was calibrated at the beginning of the campaign and routinely checked by the regional air quality network atmo Hauts-de-France".*

6. Line 168 "The yearly-mean PM10 concentration obtained for the set of sampling days was of 24.3 μg/m3". Is this the real yearly average (365 days in a year possibly measured with the beta monitor) or the average over the 122 analyzed samples? It would be great to see both values to understand if the sample selection is consistent with (indicative of, or similar to) the year average.

*The $PM_{10}$ concentration of 24.3 μg/m$^3$ reported in the manuscript corresponds to the average over the 122 analyzed samples. For the yearly average considering all the samples, the average concentration was 22.8 μg/m$^3$. The closeness of the two values is indicative that the sample selection is representative of the whole year.*

7. Figure 2. Please report the error bars relative to the bootstrap results. This would clarify some source profiles.

*Following the bootstrap results, the percentiles 25$^{th}$ and 75$^{th}$ were added to the contribution of the species by factor and were presented in Figure S2.*

[Figure]

Fig.S2: PM$_{10}$ source profiles at Cape Gris-Nez (CGN) identified using the CW-NMF model along with the percentiles 25$^{th}$ and 75$^{th}$ calculated via the bootstrap analysis.

8. Section 3.2. The description of the model setup lacks many details. In addition, it looks like you pushed the model beyond the limits of what it is capable of. It is hard to believe that you can see so many factors with just 122 samples and 28 variables without overfitting the data. For instance, the presence of a "metal-rich" factor may be due to the overfit. Thus, I would ask you to add the scaled residual plots to the SI material file along with all the diagnostics returned by the model. This is to better investigate the goodness of the selected model setup.

*As mentioned before, the authors took into consideration the comment of the referee regarding the "overfitting" of the model.*

*First of all, regarding the number of samples and species, the authors considered a total of 242 samples as input data for the application of CW-NMF model: 122 samples (1 day over three) collected in 2013 + 62 samples collected in 2013 during exceedance days and pollution episodes + 58 samples collected in 2014 at the same site. The contribution of the sources for the 122 samples (1 day over three) in 2013 were extracted from the output data of the model in order to present them in this paper.*

*Second of all, we have checked again the reconstruction of the species added into the model and came to the conclusion that three elements (La, Mn, and Cr) were not as well reconstructed as the other species. For that purpose, we had run again the CW-NMF by removing these elements. In this case, the optimal number of factors obtained was nine, removing by that the "metal-rich" source. The criteria that help us evaluate the goodness and robustness of the solution are the following:*

- *The comparison between the source profiles obtained by the model with the source profiles that can be found in the literature and in databases such as SPECIATE or SPECIEUROPE (Pernigotti et al., 2016; Simon et al., 2010) along with the evaluation of the characteristic ratios of the species in the profiles.*

- *The reconstruction of the concentration of $PM_{10}$ and all the species added into the model*

- *The bootstrap analysis that shows the variability of the solution by studying the effects from random errors and partially including the effect of rotational ambiguity. It is used to find if there is a small set of observations that can largely influence the solution. Mapping over 80% of the factors indicates that the BS uncertainties can be interpreted, and the number of factors may be appropriate.*

*In the case of the 9-factor solution, the profiles presented in figure 2 were comparable with those of the literature and the details were presented in section 3.2 of the manuscript.*

*Hereafter, we present the reconstruction of the different species included in the model as well as of $PM_{10}$ (figures will be included in the supplementary information).*

[Figure]

[Figure]

Fig. S3: Calculated species and $PM_{10}$ concentrations using Constrained Weighted - Non-Negative Matrix Factorization versus observed concentrations for all the species considered in the calculation (with the identity line as reference).

*As we can see, $PM_{10}$ and all the species were well reconstructed with slopes of the calculated vs observed linear regressions close to 1 (ranging between 0.8 and 1.11) with determination coefficient $R^2>0.9$ with the exception of Cd ($R^2=0.81$).*

*Finally, for the bootstrap analysis, the results (presented in the table below) showed values higher than 80%, indicating a stable solution and a reasonable model fit for the 9-factor solution.*

*Following all of the above-mentioned arguments, the authors do show that the 9-factor solution is robust and can be presented in the manuscript. For this reason, modifications were done in the manuscript (Section 3.3, 3.4, 3.5 as well as figures 3, 4, and 5 and the supplementary information).*

Table S1: Bootstrap mapping results for CW-NMF results at CGN site

| Bootstrap mapping Min. correlation coefficient r = 0.6 | 1 | 2 | 3 | 4 | 5 | 6 | 7 | 8 | 9 | Unmapped |
|---|---|---|---|---|---|---|---|---|---|---|
| Boot Factor 1 (Fresh sea-salts) | 100 | 0 | 0 | 0 | 0 | 0 | 0 | 0 | 0 | 0 |
| Boot Factor 2 (Aged sea-salts) | 0 | 100 | 0 | 0 | 0 | 0 | 0 | 0 | 0 | 0 |
| Boot Factor 3 (Crustal) | 0 | 0 | 99 | 0 | 0 | 0 | 0 | 0 | 0 | 1 |
| Boot Factor 4 (Secondary nitrates) | 0 | 0 | 0 | 100 | 0 | 0 | 0 | 0 | 0 | 0 |
| Boot Factor 5 (Secondary sulfates) | 0 | 0 | 0 | 0 | 100 | 0 | 0 | 0 | 0 | 0 |
| Boot Factor 6 (Biomass burning) | 0 | 0 | 0 | 0 | 0 | 99 | 0 | 0 | 0 | 1 |
| Boot Factor 7 (Road traffic) | 0 | 0 | 0 | 0 | 0 | 0 | 100 | 0 | 0 | 0 |
| Boot Factor 8 (HFO combustion) | 0 | 0 | 0 | 0 | 0 | 0 | 0 | 100 | 0 | 0 |
| Boot Factor 9 (Primary biogenic emissions) | 0 | 0 | 0 | 0 | 0 | 0 | 0 | 0 | 100 | 0 |

9. Line 227. "with an average Cl--to-Na+ ratio of 1.8 which is commonly observed for fresh sea salts (Seinfeld and Pandis, 2016)." Is the seawater composition also valid for other fresh sea salt ionic species such as K+, Mg2+, Ca2+, and SO42-?

*For the 9-factor solution that is added in the final version of the manuscript, the sea composition is also valid for the other ionic species and are resumed in the following table:*

| Ratios | $Cl^-/Na^+$ | $K^+/Na^+$ | $Ca^{2+}/Na^+$ | $SO_4^{2-}/Na^+$ | $Mg^{2+}/Na^+$ |
|---|---|---|---|---|---|
| Sea-water composition | 1.8 | 0.037 | 0.038 | 0.251 | 0.12 |
| Fresh sea-salts profile | 1.7 | 0.033 | 0.027 | 0.15 | 0.11 |

10. Line 230. Is the cations/anions ratio balanced in this factor?

*The cations/anions ratio in the "aged sea-salts" profile in the 9-factor solution is balanced with a cations/anions ratio of 1.14.*

11. Line 240-255. The authors report some literature data in support of their findings. However, most of these studies refer to finer PM (PM2.5 or even PM1). For example, the paper by Khan et al (2021) reports the EC to levoglucosan ratio in PM1. The paper by Salameh et al (2018) refers to the OC-to-EC ratio of PM2.5. It is not reliable that the ratios between different variables (chemical species) remain unchanged on PM10, PM2.5 or PM1. At least, not all. Please comment or change the references.

*The authors understand the point of view of the referee. The references were changed in order to reflect the concentration ratios between species in the $PM_{10}$ fraction. The references of Khan et al., (2021) and Salameh et al., (2018) were replaced by others that present concentration ratios in $PM_{10}$ (Amato et al., 2011; Waked et al., 2014; Sonwani et al., 2021).*

12. Figure 3. Please provide the uncertainty associated with the results.

*We agree with the referee on the importance of having uncertainty values for the sources' contributions. However, the output data of the CW-NMF model, just like the PMF model, does not include uncertainties associated with the contribution of the sources. Instead, the average concentration as well as the standard deviations of each source in ng/m³ were added in the supplementary information.*

Table S2: Average contribution and standard deviation in µg/m$^3$ and percentages of the factors to PM$_{10}$

| Source | Contrib. at CGN (µg.m$^{-3}$) | Standard deviation (µg.m$^{-3}$) | Contrib. at CGN to PM$_{10}$ (%) |
|---|---|---|---|
| Fresh sea-salts | 4.24 | 4.57 | 21.9% |
| Aged sea-salts | 3.0 | 2.31 | 15.4% |
| Crustal | 0.18 | 0.28 | 0.9% |
| Secondary nitrates | 6.27 | 9.03 | 32.3% |
| Secondary sulfates | 1.91 | 3.24 | 9.8% |
| Biomass burning | 1.49 | 3.25 | 7.7% |
| Road traffic | 0.54 | 0.61 | 2.8% |
| HFO combustion | 0.85 | 1.01 | 4.5% |
| Primary biogenic emissions | 0.92 | 1.40 | 4.7% |

13. Section 3.5. It appears that most of the factors come from the same area of origin. This again pinpoints the possible data overfit. However, the results for HFO appear to come from a more "marine" source area than the two secondary factors. It would be better to zoom in on Figure 5 to better visualize the differences. After all, we are not interested in results where the model returns too low a contribution (thus, please cut the more distant areas).

*We agree with the referee regarding his comment to figure 5. The figure was changed by replacing the contribution of the sources with the ones determined for the 9-factor solution. Additionally, the CWT representations were zoomed in order to better visualize the differences.*

[Figure]

Figure 1: CWT results for PM$_{10}$ and some NMF factors (fresh sea-salts, aged sea-salts, secondary nitrate, secondary sulfate, and HFO combustion). Red colors highlight potential emission zones. Contribution scales are in µg/m$^3$.

References:

Amato, F., Viana, M., Richard, A., Furger, M., Prévôt, A. S. H., Nava, S., Lucarelli, F., Bukowiecki, N., Alastuey, A., Reche, C., Moreno, T., Pandolfi, M., Pey, J., and Querol, X.: Size and time-resolved roadside enrichment of atmospheric particulate pollutants, Atmos. Chem. Phys., 11, 2917-2931, 10.5194/acp-11-2917-2011, 2011.

LCSQA: Suivi de l'équivalence des appareils de mesure automatique $PM_{10}$, campagnes 2011 à Metz Borny (Urbain) et Port-Saint-Louis (Industriel). 2012.

Pernigotti, D., Belis, C. A., and Spanò, L.: SPECIEUROPE: The European data base for PM source profiles, Atmos. Pollut. Res., 7, 307-314, https://doi.org/10.1016/j.apr.2015.10.007, 2016.

Simon, H., Beck, L., Bhave, P. V., Divita, F., Hsu, Y., Luecken, D., Mobley, J. D., Pouliot, G. A., Reff, A., Sarwar, G., and Strum, M.: The development and uses of EPA's SPECIATE database, Atmos. Pollut. Res., 1, 196-206, https://doi.org/10.5094/apr.2010.026, 2010.

Sonwani, S., Saxena, P., and Shukla, A.: Carbonaceous Aerosol Characterization and Their Relationship With Meteorological Parameters During Summer Monsoon and Winter Monsoon at an Industrial Region in Delhi, India, Earth and Space Science, 8, e2020EA001303, https://doi.org/10.1029/2020EA001303, 2021.

Waked, A., Favez, O., Alleman, L. Y., Piot, C., Petit, J. E., Delaunay, T., Verlinden, E., Golly, B., Besombes, J. L., Jaffrezo, J. L., and Leoz-Garziandia, E.: Source apportionment of $PM_{10}$ in a north-western Europe regional urban background site (Lens, France) using positive matrix factorization and including primary biogenic emissions, Atmos. Chem. Phys., 14, 3325-3346, https://doi.org/10.5194/acp-14-3325-2014, 2014.

---

## Author Response (AR2)

**Marc FADEL, PhD**

Unit of Environmental Chemistry and Interactions with life, UCEiV UR 4492

Université du Littoral Côte d'Opale

145 Avenue Maurice Schumann, 59140 Dunkirk, France

Email: marc.fadel@univ-littoral.fr

July 3, 2023

Dear Editor-in-Chief,

We would like to thank you for accepting our manuscript for publication in Atmospheric Chemistry and Physics. Hereafter, we have replied to the different minor comments. Amendments in the manuscript were done in track changes mode for ease of review.

1) Referee 1 comment 10 "…I do not understand why the CBPF of sea salt and aged sea salt are so different…": Please add clarifying comments to the text along the lines of that supplied in the response document.

*We have added in the manuscript additional details that were found in the response to the referee's comment (Lines 340, 341-344, 351-354 in the revised manuscript).*

*The paragraph now reads:*

*"The main reason for these differences might be the meteorological conditions as well as the seasonality. Indeed, the CBPF representation of fresh sea-salts clearly evidenced that the maximum concentrations were observed for winds blowing from the southwest (SW) and northeast (NE) wind sectors and for medium to high wind speeds (> 10 m/s). This is mainly due to the position of the sampling site, strongly subjected to fresh marine influence from sectors 210° to 50° via the North, corresponding to the English Channel and the North Sea, respectively (Error! Reference source not found.). These wind directions were predominant during all months of the year except for the summer season which could explain the higher concentrations of fresh compared to aged sea-salts. On the other hand, the maximum concentrations of aged sea-salts were obtained when the wind blew from the northeast wind sector with wind speeds higher than 10 m/s (Error! Reference source not found.). This might be explained by the reaction of the fresh sea-salts with $SO_2$ and $NO_2$ that also show the highest concentrations in the northeast wind sector (**Figure S7**) to yield aged sea-salts in the Strait of Dover and the North Sea area (Error! Reference*

*source not found.). This phenomenon occurs according to the trajectory of the air masses. By that, the aged sea-salts may not come from the wind direction open to the sea but from land (Northeast wind sector) especially in a coastal site, which is the case of this study.*

2) Referee 2 comment 2 "Please comment on the choice of sampling PM2.5…": Please add clarifying comments to the text along the lines of that supplied in the response document.

*The explanation was clearly added in the introduction of the manuscript (Lines 69-74):*

*"Several years before 2013, the EU has issued the directive 2008/50/EC that limits daily $PM_{10}$ concentrations to 50 µg/m$^3$ with a maximum of 35 days of exceedance authorized per year. The directive is binding and forces countries that do not comply with it to seek solutions for improvement. In this period, several regions in France were concerned by high number of exceedances, especially in and around Paris capital as well as in the North, the East, and the South-East parts of the country (EEA, 2014). This is why it was important to focus on the $PM_{10}$ fraction in order to understand the reasons behind these exceedances on a regional and national scale. "*

3) Referee 2 comment 6 "The yearly-mean PM10 concentrations…": Please clarify in the text that the average of the entire year is 22.8 ug/m3 and that this supports the 122 samples is representative of the entire year on average.

*The information regarding the yearly average concentration of $PM_{10}$ and the concentration for the 122 samples was added in the manuscript (Lines 182-185):*

*"The yearly mean concentration of $PM_{10}$ for the 362 samples was 22.8 µg/m$^3$ in 2013. As for the 122 samples corresponding to a sampling of 1 day over 3, the mean $PM_{10}$ is 24.3 µg/m$^3$. The closeness of the two values is indicative that the sample selection is representative of the whole year."*

4) Referee 2 comment 8 "The description of the model setup…": On line 169 (track changes version) it states that 122 samples were used as the input data, however in the response it suggests additional samples from 2013 and 2014 were also included. Please clarify the input data. If the data from the exceedance days in 2013 was included in the input data, then a discussion on if including exceedance days alters the model outputs is warranted. One can imagine that having a biased sampling (by including extra exceedance days, but not non-exceedance days) would alter the results. The 2014 data would also need explanation if that was included as input.

*We understand the editor's comment regarding the input data for the NMF model. We would like to clarify that in the response as well as in the manuscript.*

*The sampling campaign at CGN covered the year 2013 where 362 samples were collected and a part of the year 2014 (until 17/04/2014) where 107 samples were collected. Not all of these samples were chemically characterized and/or used for the source apportionment.*

*The choice was done as follows:*

- *1 day over 3 samples making a total of 122 samples in 2013 and 36 samples in 2014.*
- *51 samples in 2013 and 13 samples in 2014 corresponding to wind directions under-represented by the selection done by one day over three (non-exceedance days).*
- *11 samples with exceedances in $PM_{10}$ concentrations for 2013 and 9 for 2014.*

*This total of 242 samples was considered as the input data of the model. Besides the 1 day over 3 samples (in total 158), samples corresponding to non-exceedance days (64 samples) and to exceedance days (20 samples) were added, removing by that the bias sampling.*

*Concerning the manuscript, the authors have decided to only present the results of the sampling campaign of 2013 because it covers a whole year and by choosing the 122 samples corresponding to the 1 day over three samples. The contribution of the sources used in Figures 3,4, and 5 as well as in Figure S6 and Table S2 were extracted from the output data of the model and it only represents the 122 samples (1 day over three in 2013). The extra exceedance and non-exceedance days as well as the data of 2014 were not considered.*

*The information was clearly added in the manuscript (Section 2.2; Lines 110-116):*

*"Over the sampling period, 362 samples have been collected in 2013 and 107 in 2014. Field blanks (two per month) were also considered by placing a blank filter in sampling conditions but without pumping. Additionally, meteorological data (temperature, wind speed and direction) were recorded on site using the WMT 52 ultrasonic wind sensor (Vaisala Windcap) coupled to the DIGITEL® DA80.*

*Among these samples, 158 (122 in 2013 and 36 in 2014) corresponding to a one day over three sampling, 64 (51 in 2013 and 13 in 2014) corresponding to wind directions under-represented by the selection done by one day over three (non-exceedance days), and 20 (11 in 2013 and 9 in 2014) representing exceedance days were chosen for the chemical analysis and the source apportionment. In the results of this study, we will only be presenting the results of the 122 samples representing a sampling of one day over three in 2013."*

Additionally, the bootstrap analysis would benefit from a few additional sentences of explanation (SI is ok) and the results of Table S1 require a few sentences explanation of the meaning of the table (where is the number 100 coming from?).

*A description of the bootstrap analysis method as well as details regarding the number 100 was added in the supplementary information:*

*"The Bootstrap analysis show the effects from random errors and include partially the effects of rotational ambiguity. It is used to find if there is a small set of observations that can largely influence the solution. This method creates sets of bootstrap data constructed by randomly selecting blocks of observations from the initial dataset. The size of the block was taken as 5 samples for this study. The solution was bootstrapped 100 times to ensure the robustness of the results. Mapping over 80% of the factors indicates that the bootstrap uncertainties can be interpreted, and the number of factors may be appropriate. "*

*The interpretation of the results of Table S1 was added in the manuscript as follows (Lines 236-238):*

*"The best results were obtained for the 9 factors solution. The stability of the results was examined via bootstrap analysis and the different source profiles satisfied the validation criterion (**Table**

*S1) with a mapping percentage of at least 99% (higher than 80%) showing the robustness of the obtained solution".*

5) Referee 2 #9 "Is the seawater composition…": Please add a sentence to the text saying that the ratios were also valid for other ionic species.

*The validity of the other concentration ratios was added into the manuscript (Lines 250-251):*

*"The concentration ratios between the other ionic species were also valid for sea water composition: $K^+/Na^+=0.03$, $Ca^{2+}/Na^+=0.03$, $SO_4^{2-}/Na^+=0.15$, and $Mg^{2+}/Na^+=0.11$."*

6) Referee 2 #10 "Is the cations/anions ratio balanced in this factor?": Please clarify in the *text that it has a ratio of 1.14.*

*The information was added in the manuscript as follows (Line 255):*

*The ionic balance is respected in this factor with a cations/anions ratio of 1.14.*

7) Please ensure that the SI material is cited in order. Currently Fig. S3 is cited before Fig. S2.

*We have carefully checked the manuscript and changed the order of the figures in the supplementary in order to cite them in order.*

8) Fig. S6: Is the 75th percentile of the January value for secondary sulfates off-scale? It is hard to tell with the current axis.

*We would like to thank the editor for the comment. The axis for the figure of secondary sulfates (Fig S6) was modified and added in the SI.*

References:

EEA: Air quality in Europe 2014 report, EEA report, Publications Office of the European Union, Luxembourg2014.
Ledoux, F., Roche, C., Cazier, F., Beaugard, C., and Courcot, D.: Influence of ship emissions on $NO_x$, $SO_2$, $O_3$ and PM concentrations in a North-Sea harbor in France, J. Environ. Sci., 71, 56-66, https://doi.org/10.1016/j.jes.2018.03.030, 2018.